# Identifying the diabatic processes driving the evolution of a sting jet: the case of Storm Ciarán

Ambrogio Volonté<sup>1,2</sup>, Hanna Joos<sup>3</sup>, Ming Hon Franco Lee<sup>3</sup>, Richard Forbes<sup>4</sup>, and Rémi Bouffet-Klein<sup>3</sup>

Correspondence: Ambrogio Volonté (a.volonte@reading.ac.uk), Hanna Joos (hanna.joos@env.ethz.ch)

## Abstract.

Sting jets (SJs) are airstreams that can lead to exceptionally strong and damaging winds in intense extratropical cyclones. Whilst there is extensive evidence that SJ descent can be associated with the release of symmetric instability (SI), the individual diabatic processes driving the onset of this instability have not yet been identified and characterised. In our study we tackle this question by analysing a near-operational IFS simulation of Storm Ciarán, that featured a SJ associated with damaging winds and characterised by the development of SI (indicated by negative potential vorticity, PV) during its evolution. Diabatic tendencies are included in the output of this simulation and are used in our study, including by being traced on Lagrangian trajectories, to illustrate the contributions of the individual diabatic processes to the onset of SI during the evolution of the SJ.

The SJ in our simulation is consistent in terms of magnitude, timing and structure with operational forecasts, observations and literature. This SJ develops in an environment characterised by multiple bands of negative PV in the cloud head. It becomes part of one of such filaments as it ascends near the bent-back warm front before descending off the tip of the cloud head. The decrease in PV observed on the SJ trajectories at this time is only partially captured by diabatic tendencies. This large discrepancy exposes the limitations of the methodology and can be ascribed to the use of offline trajectories computed from hourly instantaneous model data in an environment characterised by fast-changing and non-linear processes and by fully three-dimensional and small-scale patterns. This is particularly true in the narrow region near the bent-back warm front and the tip of the cloud head, in which the SJ travels as SI develops along it.

The decrease in PV along the SJ captured by diabatic tendencies is mainly associated with four moist processes: condensation of water vapour, evaporation of cloud water, melting of ice and snow and sublimation of snow. The first three show large variability across trajectories and, particularly for condensation, positive and negative extremes near the trajectories. The small decrease in PV caused by the sublimation of snow is instead consistent across trajectories. The reduction in buoyancy caused by the cooling from snow sublimation and melting favours the start of SJ descent, rather than the continuation of ascent.

In summary, in this study we analyse diabatic tendencies in a model simulation of Storm Ciarán, acknowledging their limitations, to reveal the role of different moist processes in causing the onset of instability on a SJ and therefore driving its intensification. The complex interplay between these processes highlights the unique properties of the narrow region in the cloud head in which the SJ develops before descending towards the surface and bringing damaging winds.

<sup>&</sup>lt;sup>1</sup>Department of Meteorology, University of Reading, Reading, UK

<sup>&</sup>lt;sup>2</sup>National Centre for Atmospheric Science, University of Reading, Reading, UK

<sup>&</sup>lt;sup>3</sup>Institute for Atmospheric and Climate Science, ETH Zurich, Zurich, Switzerland

<sup>&</sup>lt;sup>4</sup>European Centre for Medium-Range Weather Forecasts, Reading, United Kingdom

## 1 Introduction



## 1.1 Sting jets in extratropical cyclones

Extratropical cyclones are a primary cause of severe impacts in the mid-latitudes, including over the UK and Europe, repeatedly causing extensive damage and loss of life due to extreme winds and precipitation. Strong winds in extratropical cyclones can be linked to a number of distinct airstreams, often long-lived and of synoptic-scale size, such as the widely studied warm (Browning, 1971) and cold conveyor belts (Schultz, 2001) and the descending dry intrusion (Browning and Roberts, 1994). However, in Shapiro-Keyser cyclones (Shapiro and Keyser, 1990) strong surface winds can also be associated with a shorter-lived and smaller-scale airstream, called "sting jet". A sting jet (SJ) is a mesoscale airflow that descends off the tip of the cyclone cloud head and accelerates into its frontal-fracture region. Despite their local and transient nature, SJs can lead to distinct regions of exceptionally strong and damaging near-surface winds. First identified two decades ago when re-examining the Great Storm that devastated southern England on 16 October 1987 (Browning and Field, 2004; Clark et al., 2005), SJs were since identified in some of the most damaging cyclones affecting the UK and Europe (Clark and Gray, 2018). Most recent examples of notable SJ cyclones include:

- Storm Eunice (Volonté et al., 2023a, b), which hit the British Isles on 18 February 2022, triggering the first UK Met Office red warning for wind over southeast England since the alert system was set up in the wake of the 1987 Great Storm. Storm Eunice caused extensive damage and broke the gust speed record for England with 106 knots recorded at Needles Old Battery (admittedly an exposed site, off the west coast of the Isle of Wight).
  - Storm Ciarán (Volonté and Riboldi, 2024; Gray and Volonté, 2024; Charlton-Perez et al., 2024), a highly diabatic wind-storm that hit the UK and western Europe on 1-2 November 2023 causing multiple fatalities and severe damage, including the loss of electricity for 1.2 million people in France. Reported gusts exceeded 110 knots in Brittany and Jersey (Channel Islands) experienced a destructive tornado, likely the strongest reported in the British Isles since 1954.
    - Storm Éowyn (Kendon, Mike, 2025; Suri et al., 2025), that impacted the British Isles on 24 January 2025 after undergoing a central pressure drop of 50hPa in 24 hours, i.e., more than twice the rate in the definition of explosive deepening (Sanders and Gyakum, 1980). Red weather warning were issued by the Met Office and Met Éireann, as Storm Éowyn was one of the most severe windstorms to affect the region in recent years, setting a provisional gust speed record for Ireland (100kn at Mace Head, Galway) and causing severe infrastructure damage over Ireland and the northern parts of the UK.



# 1.2 The roles of symmetric instability and of moist processes in the dynamics of SJs

The three storms just listed have all in common the development of a SJ that is associated with the onset of symmetric instability (SI) in the cloud head, as illustrated in Volonté et al. (2023a) for Storm Eunice, Gray and Volonté (2024) for Storm Ciarán and by preliminary analysis of Met Office operational forecast data for Storm Éowyn (not shown). As explained in detail in Clark and Gray (2018), SI is essentially a form of inertial instability generalised to a baroclinic flow. It is indicated by negative values of Ertel potential vorticity (PV hereafter), which is defined as PV = \(\frac{\xi}{\rho}\theta\), where \(\rho\) is the air density, \(\xi\)
the absolute vorticity vector, and \(\theta\) the potential temperature. SI is derived applying the semi-geostrophic approximation to a two-dimensional base state that is in thermal wind balance and has a uniform thermal gradient. SI is released as a consequence of slantwise displacements happening at angles between the slopes of absolute geostrophic momentum and \(\theta\). The moist counterpart of SI is conditional symmetric instability (CSI), that is defined by replacing \(\theta\) with saturated equivalent potential temperature (\(\theta\_e^\*\)). As surfaces of \(\theta\_e^\*\) are more sloped than those of \(\theta\), CSI is more common than SI. However, CSI requires saturation to be released, while SI does not.

The hypothesis that the acceleration and descent of SJs could be at least partly associated with the release of CSI or even SI dates back to the pioneering works of Browning (2004); Browning and Field (2004). In their analysis of the Great Storm, they highlighted the banded structure of the cloud-head tip and illustrated it as an indication of multiple slantwise circulations, with descending cloud-free branches associated with the acceleration of the sting jet out of the cloud head. The associated seminal modelling work by Clark et al. (2005) confirmed the presence of these slantwise circulations, with the SJ appearing within their descending part. Subsequent studies added further evidence to the hypothesis that SJ descent and acceleration is associated with slantwise circulations caused by mesoscale instability in cloud-head frontal circulations (Gray et al., 2011; Martínez-Alvarado et al., 2014) or at least reduced or neutral stability, as in Coronel et al. (2016). The review by Clark and Gray (2018) stresses the likelihood of a continuum of behaviours, "from balanced descent partly associated with frontolysis in the frontal-fracture region, through horizontally smaller scale and stronger frontolytic descent associated with weak stability to slantwise convective downdraughts, to multiple slantwise convective downdraughts associated with the release of CSI and even, possibly, SI." They also highlight the evidence that this release of instability is associated with substantial speed-up on top of that provided by the quasi-geostrophic frontolytic descent and in a region of the cyclone usually already prone to strong winds.

Volonté et al. (2018) used model simulations of Storm Tini to develop a conceptual model illustrating the primary role of vorticity tilting via slantwise highlighting also its close link to negative PV. This conceptual model was confirmed by the idealised simulations in Volonté et al. (2020), showing bands of strongly positive and negative horizontal vorticity being associated with tight frontal circulations in the cloud head. Large negative values of the tilting term in the vertical vorticity budget were diagnosed along SJ trajectories travelling towards the tip of the cloud head, where the formation of a narrow filament of negative PV just outside the bent-back warm front was observed. This formation of negative PV filaments along the SJ as it travels in a narrow region in the cloud head and near the bent-back front has also been confirmed by the most recent




studies of intense sting jet storms, such as Eunice and Ciarán, as explained in Section 1.1 (Volonté et al., 2023b; Gray and Volonté, 2024). It can thus confidently be stated that intense SJ are normally characterised by the presence of SI.

SJ cyclones are often characterised by banding in the cloud head (Clark and Gray, 2018), although caution might be exerted, as discussed in Schultz and Schumacher (1999), in associating it to slantwise circulations and mesoscale instability. Slantwise circulations in the cloud head have been observed in rapidly developing cyclones, such as the FASTEX IOP16 cyclone described in Roberts and Forbes (2002) and Lean and Clark (2003), in which the strength of the circulations led to describing them as forming "multiple cloud heads." Clough and Franks (1991) highlighted the importance of sublimation of ice as an effective mechanism for the maintenance of slantwise descent, thanks to the associated cooling. The role of latent cooling caused by precipitation (liquid or solid) falling from above and helping to maintain descending motions is stressed also by Forbes and Clark (2003). Focusing on SJs, Clark et al. (2005) found a strong correlation between evaporative cooling and SJ descent. Subsequent studies provided differing results on the importance of latent cooling for the descent and acceleration of SJs (Clark and Gray, 2018). It is worth noting that, given that the presence of a precipitating cloud above the airstream is a necessary condition for the cooling to take place on it, these processes become less likely as the SJ progresses in its journey off the tip of the cloud head and into the frontal-fracture region.

In summary, the link between SJ dynamics and the presence of SI, or of reduced stability, has now been established. The onset of this instability, indicated by negative PV, is driven by tight frontal circulations in the cloud head, near the bent-back warm front. This suggests the likely involvement of moist processes, that can help to initiate and maintain the descent via latent cooling. However, the distinct roles of individual diabatic processes in reducing PV along the SJ as it travels in the cloud head, before starting its descent and acceleration out of it, have not been identified yet.

# 1.3 Identifying individual diabatic processes in extratropical cyclones

In the last couple of decades two methods have been mainly used to investigate the role of individual diabatic processes in the evolution of extratropical cyclones and associated features, such as SJs, assessing the related changes in PV and/or potential temperature ( $\theta$ ). These are passive tracers and tendency diagnostics. We summarise the two approaches here, while a more comprehensive description, also featuring a variety of examples from the literature, can be found in Section 5.4 of Wernli and Gray (2024).

The use of tracers follows from the seminal work by Stoelinga (1996), in which the accumulation of PV increments caused by the different non-conservative model processes was partitioned using the routines and parameterisation schemes of the model and advected at every time step, together with the conserved component, using the model advection scheme. Since then, most diabatic tracers studies have been performed using the Met Office Unified Model (MetUM). Particularly relevant for our study are the works by Chagnon et al. (2013) and Chagnon and Gray (2015), focusing on the role of different diabatic processes in modifying PV in several regions of extratropical cyclones. The requirement of tracers to be incorporated in the model code represents an obstacle towards easy use and portability. After around three decades since pioneering works, studies implementing diabatic tracers in models other than the MetUM are still limited, one example being the analysis by Flaounas et al. (2021) of diabatic and baroclinic contributions to the evolution of intense Mediterranean cyclones using the WRF model.


An alternative approach is the use of tendency diagnostics. In this case tendencies from the different diabatic and frictional processes are included in the model output. These tendencies are then traced along trajectories, normally computed offline using the resolved wind. This is what was done by Joos and Wernli (2012), who showed that it was possible to use the heating tendencies available in the output of the COSMO model to provide a detailed breakdown of the role of individual diabatic processes in modifying PV in a warm conveyor belt. This study was followed by further work using diabatic tendencies to reveal the role of non-conservative processes on the structure and evolution of warm conveyor belts using the IFS model (Joos and Forbes, 2016) and, more recently, the ICON model (Oertel et al., 2023). Other main features of extratropical weather have also recently been analysed using diabatic tendencies in the IFS, such as the jet stream and the tropopause (Spreitzer et al., 2019) and, of particular relevance to our study, tropospheric fronts in extratropical cyclones (Attinger et al., 2019, 2021). In Attinger et al. (2019), the analysis of diabatic tendencies in an extratropical cyclone case study showed that PV increase along the warm front was mainly generated by condensation and turbulence, while PV decrease was primarily caused by snow melting and sublimation. Focusing on the bent-back end of the warm front and the cyclone centre, condensation, convection, snow melting and sublimation were found to be the most important contributors to PV increases. In Attinger et al. (2021), the same approach was extended to a series of month-long global free-running simulations, providing a systematic analysis of low-level PV modification along the different fronts characterising extratropical cyclones. This analysis confirms that at the warm front high PV is mainly caused by condensation (in the cold season) and turbulent mixing of momentum (in the warm season) while melting and snow sublimation play a role in the generation of low PV, particularly to the north of the front. At the bent-back front and at the cyclone centre, condensation is still the main contributor to PV increase.

Both the passive tracers and tendency diagnostics methods are based on model parameterisation schemes. Both methods also highlight the Lagrangian nature of modifications to PV and  $\theta$ , as non-conservative processes are integrated (online or offline) along the flow. For this study we choose to use the tendency diagnostics methods, so that we can take advantage of an established set up that has already proven to be effective for the identification of individual diabatic processes when applied on fronts and other features characterising extratropical cyclones.

# 1.4 Our plan


Therefore, in this study we follow the same diabatic tendency methodology as in Attinger et al. (2021) and we apply it to a model simulation of Storm Ciarán. As described at the beginning of this introduction, Ciarán was a highly diabatic high-impact windstorm that featured a SJ displaying a clear SI signal (Gray and Volonté, 2024). Hence, it is an appropriate case study to investigate the role of diabatic processes in the evolution of SJs and the related onset of SI.

The remainder of this article is structured as follows. The model simulation, use of diabatic tendencies and tracing along Lagrangian trajectories are described in Section 2. The results section (Section 3) begins with the illustration of the presence of a symmetrically unstable SJ and a description of the evolution of PV along SJ trajectories. This is followed by a detailed analysis of the diabatic processes modifying PV along the trajectories and in the environment around them. Section 4 completes the article as it contains a discussion of the results and the overall conclusions.




## 2 Data and methods

## 2.1 IFS model and use of diabatic tendencies

The simulation analysed in this study is performed using the Cycle 47r3 version of the Integrated Forecasting System (IFS) model of the European Centre for Medium-Range Weather Forecasts (ECMWF), which was operational from October 2021 to June 2023 at ECMWF. A 2-day forecast is initialised with ECMWF analysis field at 00UTC on 01 November 2023 and it is run at a cubic-octahedral spectral transform discretisation of TCo1279 with 137 vertical levels, the same as the operational forecast. The corresponding horizontal resolution is approximately 9 km and the hourly output fields are interpolated to a regular grid at  $0.1^{\circ}$  resolution.

Apart from the standard model outputs, the temperature, moisture, and momentum tendencies associated with individual parametrised subgrid-scale processes are retrieved, similar to Spreitzer et al. (2019); Attinger et al. (2019, 2021). Using these tendencies, the PV framework is applied to evaluate the effect of diabatic processes on the formation of sting jets. As PV is materially conserved under adiabatic and frictionless flow, any changes in PV can be linked to potential temperature ( $Q_i$ ) and momentum tendencies ( $F_i$ ) imposed by the non-conservative processes i, shown in table 1. The material change rate of PV (PVR) can be described by

$$PVR_i = \frac{1}{\rho} (\boldsymbol{\xi} \cdot \nabla Q_i + \nabla \times \boldsymbol{F_i} \cdot \nabla \theta)$$
 (1)

where again  $\rho$  is the air density,  $\xi$  the absolute vorticity vector, and  $\theta$  the potential temperature. Compared to Attinger et al. (2021), the cloud microphysical processes considered in this study are slightly modified, with a major update of the moist physics (Bechtold et al., 2020) applied on Cycle 47r3 of the IFS. In particular, melting of ice and snow is grouped into a single process "melting" and the freezing and riming of cloud droplets are grouped into a single process "freezing", as detailed in Table 1.

## 2.2 Lagrangian perspective on PV modification

In order to evaluate the influence of diabatic processes on the evolution of the sting jet, backward trajectories are calculated with the Lagrangian Analysis Tool LAGRANTO (Sprenger and Wernli, 2015). The starting region of the backward trajectories is described in detail in section 3.1.2. All momentum and potential temperature tendencies are traced along the trajectories of interest. With this approach we can quantify the instantaneous rate of change in PV due to every process separately (PVR) and then calculate the time-accumulated PV tendency (APV) along the flow. The net PV modification is described by the sum of large-scale microphysics, radiation, convection and turbulence, plus a residual term:

$$\Delta PV = APV_{all} + RES = APV_{cloud} + APV_{sw} + APV_{lw} + APV_{conv} + APV_{turb} + RES \tag{2}$$

| PV tendency                              |                               | Process                                                                | Tendency           |
|------------------------------------------|-------------------------------|------------------------------------------------------------------------|--------------------|
| instantaneous                            | accumulated                   |                                                                        |                    |
| $PVR_{sw}$                               | $APV_{sw}$                    | Short-wave radiation                                                   | $Q_{sw}$           |
| $PVR_{\mathrm{lw}} \\$                   | $APV_{\mathrm{lw}} \\$        | Long-wave radiation                                                    | $Q_{\mathrm{lw}}$  |
| $PVR_{turb} \\$                          | $APV_{turb} \\$               | Turbulence, orographic and non-orographic gravity wave drag            | $Q_{turb}$         |
| $PVR_{turb\mathit{F}}$                   | $APV_{turbF}$                 | Turbulence, orographic and non-orographic gravity wave drag (momentum) | $F_{turb}$         |
| $PVR_{conv} \\$                          | $APV_{conv}$                  | Convection                                                             | $Q_{conv}$         |
| $\mathrm{PVR}_{\mathrm{conv}\mathit{F}}$ | $APV_{convF}$                 | Convection (momentum)                                                  | $F_{conv}$         |
| $PVR_{cloud} \\$                         | $APV_{cloud}$                 | Cloud microphysics                                                     | $Q_{cloud}$        |
| $PVR_{cond}$                             | $APV_{cond}$                  | Condensation of water vapour                                           | $Q_{cond}$         |
| $PVR_{evc} \\$                           | $APV_{evc}$                   | Evaporation of cloud water                                             | $Q_{evc}$          |
| $PVR_{evr} \\$                           | $APV_{evr}$                   | Evaporation of rain                                                    | $Q_{evr}$          |
| $PVR_{dep}$                              | $APV_{dep}$                   | Depositional growth of ice                                             | $Q_{dep}$          |
| $PVR_{subi} \\$                          | $APV_{subi}$                  | Sublimation of ice                                                     | $Q_{subi}$         |
| $PVR_{subs} \\$                          | $APV_{subs}$                  | Sublimation of snow                                                    | $Q_{subs}$         |
| $PVR_{melt} \\$                          | $APV_{melt} \\$               | Melting of ice and snow                                                | $Q_{melt}$         |
| $PVR_{frz}$                              | $\mathrm{APV}_{\mathrm{frz}}$ | Freezing of cloud water and rain                                       | $Q_{\mathrm{frz}}$ |

**Table 1.** Abbreviations and description of the non-conservative processes studied. PVR and APV stand for instantaneous and time-accumulated PV tendency, respectively, while Q stands for instantaneous potential temperature tendency and F for instantaneous momentum tendency. All PV tendencies are based on  $\theta$  tendencies unless explicitly mentioned.

where *RES* denotes the residual. This Lagrangian method of tracing diabatic tendencies along trajectories has already been successfully applied to the formation of PV anomalies in warm conveyor belts (Joos and Wernli, 2012; Joos and Forbes, 2016), in extratropical cyclones (Attinger et al., 2019, 2021), and at the tropopause level (Spreitzer et al., 2019). A more detailed description of the method can also be found in (Spreitzer et al., 2019) and (Attinger et al., 2019).

## 3 Results

# 3.1 Evidence of a SJ associated with symmetric instability

Storm Ciarán was, as described in the introduction, a severe and highly diabatic windstorm that featured SJ development. In this section we show that a clear SJ is identified in the IFS simulation performed and used in this study, illustrate its properties and highlight that this SJ is associated with SI.



## 3.1.1 Eulerian overview

**Figure 1.** a) Wind speed (every 2.5 ms<sup>-1</sup>, shading from 40 ms<sup>-1</sup> to 55 ms<sup>-1</sup> and green dashed contours above 55 ms<sup>-1</sup>) and equivalent potential temperature ( $\theta_e$ , magenta contours, K)) at 850 hPa, relative humidity at 700 hPa (grey shading, > 80%) and mean sea level pressure (black contours) at 15UTC on 01 November. b) As in a) but on the vertical cross-section along transect AB.

Figure 1 contains a map (Fig. 1a) that shows Storm Ciarán at 15UTC on 1 November 2023, the time at which the highest wind speed at 850hPa was simulated. At this time, Ciarán is undergoing explosive deepening (not shown) whilst developing into a severe Shapiro-Keyser extratropical cyclone. This is indicated by the presence of a bent-back warm front wrapping around the cyclone centre (see equivalent potential temperature,  $\theta_e$ , contours) and by the south-eastward displacement of the cold front, leading to the opening of a frontal-fracture region in between the two fronts, just south of the cyclone centre. Given the fast eastward movement of Storm Ciarán, is not surprising to see that the highest wind speeds at 850hPa are found on the southern side of the cyclone, where cyclonic circulation and system translation add up. The maximum values, substantially higher than those near the cold front and in the warm sector, are limited to a small and distinct area in the frontal-fracture region, just off the tip of the cloud head. This is where SJs are expected to be located at the end of their descent and acceleration.

A vertical cross-section cutting through the wind maximum in the frontal-fracture region (Fig. 1b) can be used together with the map just described to understand the vertical structure of the cyclone near the tip of the cloud head, across the frontal-fracture region, the cold front and warm sector. The low-level wind maximum in the frontal-fracture region, characterised by values nearing 55 ms<sup>-1</sup> is centred at around 850 hPa and located just outside the tip of the cloud head and at the bottom of an area of downward moist isentropes, consistent with the presence of a descending SJ airstream. Above it, wind values close to  $50 \text{ ms}^{-1}$  are found at around 600 hPa near a local minimum in  $\theta_e$ , pointing at the presence of the dry intrusion, with the core of the upper-level jet, exceeding  $60 \text{ ms}^{-1}$  near 300 hPa, above it. Winds exceeding  $40 \text{ ms}^{-1}$  at low-levels are found at low-levels at either side of the frontal-fracture region, with the area above  $45 \text{ms}^{-1}$  in the warm sector, ahead of the cold front, indicating





the warm conveyor belt, and the small local maximum just above  $40 \text{ms}^{-1}$  on the cold side of the bent-back front showing the front-edge of the cold conveyor belt.

The three-dimensional structure and timing of the airstreams just described are consistent with those displayed in the Met Office operational forecasts illustrated in Gray and Volonté (2024) and also with the satellite imagery presented there. This provides the necessary confidence that the IFS simulation analysed in this study is a realistic representation of Storm Ciarán, justifying its use when investigating SJ dynamics.

## 215 3.1.2 Lagrangian trajectories

Backward trajectories are computed from the wind speed maximum in the frontal-fracture region that indicates the SJ. The constraints applied to these trajectories, listed below, are consistent with the general evolution of SJs associated with the onset of SI (i.e., negative PV) and in particular with the properties of the SJs identified in the simulations of Storm Ciáran presented in (Gray and Volonté, 2024), such as the relatively small descent and the quick ascent/descent pattern.

In detail, the horizontal boundaries of the initial domain are 15°W, 12.5°W, 46.5°N and °48N (with a 0.1° grid spacing) and the vertical extent goes from 950 hPa to 700 hPa (every 10 hPa). The release time is 15UTC on 1 November, when the low-level wind speed reaches its highest maximum value, and are computed backwards for 15 hours. Trajectories are retained if meeting the following criteria:

- wind speed exceeding 52 m/s at release time;
- pressure increase (i.e., descent) exceeding 50hPa in the three hours before the release time;
- negative PV four hours before the release time.

Hence, while 10816 trajectories are initially calculated, the number of those that are retained decreases to 195, 113 and then down to 51 as those three criteria are applied in succession. These 51 trajectories are all associated with a symmetrically unstable SJ producing the highest low-level wind speed present in our IFS simulation of Storm Ciarán. The time evolution of their key properties is illustrated in Figures 2 and 3.

Figure 2 contains time profiles of wind speed, pressure, PV and relative humidity (RH) with respect to water along the selected SJ trajectories. Fig. 2a shows that all the acceleration along the trajectories occurs in the latest five hours (i.e., from 10UTC), when the airstream starts being aligned with the storm propagation and accelerates relative to the cyclone centre (not shown). Fig. 2b shows that most of this increase in wind speed happens when the SJ is descending. The trajectories stay in the boundary layer, with pressure close to 900 hPa until 7 hours before the release time (t=-7h, 08UTC), they then ascend up to around 765 hPa at t=-4h (11UTC), before descending towards 850hPa.

Fig. 2c illustrates the evolution of PV along the trajectories, with values initially slightly above zero and a temporary increase to just above 0.5 PVU at t=-6h (09UTC). This is followed by a rapid decrease, with the mean and median PV values reaching around -0.5 PVU by t=-4h (11UTC), i.e. by the end of the ascent. As PV returns back towards zero during the final descent, variability across trajectories increases substantially, with more than 2 PVU separating the  $5^{th}$  and  $95^{th}$  PV percentiles at



**Figure 2.** Time evolution of (a) wind speed (m s<sup>-1</sup>), (b) pressure (hPa), (c) potential vorticity (PVU), and (d) relative humidity with respect to water (%) along SJ backward trajectories (see details of identification criteria in the text). Black and green solid lines in these panels indicate the median and mean of each field, respectively. Dark grey shading covers the values between the 25<sup>th</sup> and 75<sup>th</sup> percentiles and light grey shading covers the values between the 5<sup>th</sup> and 95<sup>th</sup> percentiles. Percentiles are calculated independently at each time, hence a single percentile line can refer to different trajectories at different times.

release time. Looking at Fig. 2d we can see that RH hovers around 85-90% in the hours before the SJ descent, with a minor increase above 90% peaking at t=-5h (10UTC). This is followed by a rapid drying up during the descent with values decreasing sharply towards 50%. In summary, Figure 2 highlights the typical SJ behaviour of the selected trajectories, displaying at first the onset of SI during the ascent in near-saturated air and then its release as the airstream descends, accelerates and dries up.

Figure 3 allows us to have a closer look at the moist processes associated with the SJ evolution just described. Fig. 3a highlights the marked increase in potential temperature,  $\theta$ , from around 288K to 294K, that characterises the ascent of the SJ, from t=-6h to t=-4h (09UTC to 11UTC). There is instead minimal change in  $\theta$  during the following descent, which can thus be characterised as mostly adiabatic. The increase in dry  $\theta$  during the ascent is not mirrored by its equivalent counterpart ( $\theta_e$ , not shown), hence indicating that it is primarily driven by diabatic processes acting on the trajectories rather than mixing with warmer air. This diabatic warming is associated with a clear decrease of around 1.5-2 g kg<sup>-1</sup> in specific humidity, displayed in Fig. 3b, and, initially, a slight increase in RH. This suggests the occurrence of condensation. Changes in specific humidity during the final descent are negligible, confirming its predominantly adiabatic character.

Fig. 3c contains time series of cloud liquid water content (black line and grey bands) and rain water content (blue line and bands). There is a small and near-constant rain water content on the trajectories while they travel in the low levels, then becoming negligible, but with increased variability as the airstream ascends, between t=-7h and t=-4h. At the same time, the


Figure 3. As Figure 2 but for (a)  $\theta$  (K), (b) specific humidity (g kg<sup>-1</sup>), (c) rain and liquid water content (black and blue, respectively, g kg<sup>-1</sup>) and (d) snow water content (g kg<sup>-1</sup>).

variability across trajectories of liquid water content, on average close to zero, also increases. This enhanced variability with little change in mean and median values suggests that water content in different species is being added and removed from the trajectories at similar rates, for instance by condensation and rain steadily acting during the SJ ascent. Snow water content is displayed separately in Fig. 3d. Snow water content is equal to zero for all trajectories until the start of the ascent at t=-7h. It then increases abruptly, with the median,  $75^{th}$  and  $95^{th}$  percentiles reaching  $0.8g \text{ kg}^{-1}$ ,  $1.2g \text{ kg}^{-1}$  and  $1.6g \text{ kg}^{-1}$  and at t=-5, respectively. These values are substantially higher than the  $0.3 - 0.4g \text{ kg}^{-1}$  peak  $95^{th}$  percentiles reached by rain and liquid water content. This large increase in snow water content happens as trajectories ascend towards the freezing level, with mean and median temperature values decreasing from around  $5^{\circ}\text{C}$  ( 278K) at t=-7h to just below zero at t=-4h, before warming again during the final descent (Figure A1a). This behaviour thus suggests that substantial snow is falling into the airstream.

To summarise, this section shows that the selected trajectories display a behaviour that is typical of a SJ associated with SI and that the onset of this instability as the airstream ascends is characterised by the joint action of liquid and solid moist processes, with snow becoming more important as the SJ moves closer to the freezing level. To better understand how these processes drive the dynamics of the SJ, in the next sections we first analyse the three-dimensional evolution of PV and then identify the diabatic processes (starting from the indications from the profiles just described) playing a role in changing PV.

## 270 3.2 PV evolution along SJ trajectories

In this section we use the horizontal maps in Figure 4 and vertical cross-sections in Figure 5 to detail the evolution of PV along SJ trajectories and in the environment around them, focusing in particular on the cloud head and the bent-back warm front.

The different panels of Figure 4 cover the evolution of the rapidly deepening cyclone from before the start of the SJ ascent to the end of its descent. At each of these times they illustrate the structure of the storm at the pressure level that is nearest to the mean pressure of the trajectories. The two vertical cross-sections in Figure 5 are generated using the transects in Figure 4e. They refer to 11UTC, the end of the SJ ascent when PV has decreased below zero on all trajectories. They are orientated along and across the SJ and the negative PV filament associated with it.

**Figure 4.** PV (shading) and RH with respect to water (grey shading, > 80%) at the pressure levels indicated and mean sea level pressure (dashed black contours) at the times indicated. Green dots show the location of trajectories at the related times. In each panel the pressure level chosen is the closest to the mean pressure of the trajectories.






Figs. 4a-b refer to when the SJ trajectories are yet to start ascending, before 09UTC, and are found at low levels on the outer (i.e. cold) side of the warm front, with widespread low or negative PV around them. Figs. 4c-e show that as the SJ trajectories ascend up to above 775 hPa by 11UTC, the cyclone continues to evolve following the Shapiro-Keyser conceptual model, with the cold front detached from the bending-back warm front. The SJ trajectories rotate cyclonically around the warm front and at this time are located at the rear end of a narrow filament of negative PV found just on the cold side of the front and roughly parallel to it. The structure and location of this filament is consistent with the results from the idealised simulations shown in Volonté et al. (2020) and thus with the conceptual model of SJ evolution confirmed in that study, after its development in Volonté et al. (2018). The front end of this filament is near an opening at the tip of the cloud head, with a small negative-PV region separated from the main filament and already off the cloud. This suggests the presence of earlier SJ activity in the cloud head, in the same way as the presence of another negative-PV filament behind the trajectories points at the potential for later sting-jet activity (although in practice this might be prevented by the evolution of the cyclone towards a mature stage and the closing of the frontal-fracture region). Additional weaker high-low PV bands (with some pockets of negative PV) are present to the outside of the warm front and the SJ negative-PV filament. These bands, also oriented along the front and sitting within the cloud head, suggest the presence of multiple secondary circulations around the main frontal zone and indicate potential for SI even in outer regions.

Figs. 4f-g show the SJ trajectories travelling as a compact group in the initial stages of their descent while leaving the tip of the cloud head. At 13UTC the negative-PV region in which they lie is almost exactly co-located with the cloud gap characterising the tip of the cloud head, further confirming that descent, with its consequent decrease in relative humidity, is taking place around the trajectories. Figs. 4h-i bring us to 15UTC, the time of release of these backward trajectories. The bent-back front is continuing its wrapping process and the trajectories are now ahead of the tip and descended into the frontal-fracture region, where they are reaching their maximum speed, as shown by the wind speed profile in Figure 2 and the map and vertical cross-section in Figure 1). There is now another negative-PV filament that is reaching the tip of the cloud head and opening a gap in the cloud head as it presumably descends and accelerates as a later SJ.

There is thus plenty of evidence that SJ activity took place for an extended period of time, with multiple pulses of unstable descending air. The trajectories analysed in this study are part of the SJ pulse that produced the highest wind speed at 850 hPa in this IFS simulation as it occurred at the most favourable time during cyclone evolution and took full advantage of environmental conditions.

The two vertical cross-sections in Figure 5 display the environment around the SJ trajectories at 11 UTC, the end of the ascent. Section AB, in Fig. 5a, is oriented along the negative-PV filament and shows that the trajectories are found in an elongated area of negative PV centred between 700 hPa and 800 hPa. Whilst the SJ trajectories are still in the final stages of their ascent, the negative-PV filament is generally oriented downward, with some regions extending down towards the surface. The area of markedly positive PV, with widespread values exceeding 3 PVU above the negative-PV filament is part of the outward-slanted bent-back warm front, intercepted by the section between 600 hPa and 700 hPa. Section CD, in Fig. 5b, is oriented across the bent-back warm front and the negative-PV filament. It confirms the slanted structure of the warm front from around 750 hPa upwards, whilst the intense front is upright at lower levels. The same slanted structure is displayed by the



Figure 5. Vertical cross-sections of PV (shading), RH with respect to water (grey shading, > 80%) and  $\theta_e$  (magenta contours) along the (a) AB and (b) CD transects shown in Figure 4e, referring to 11 UTC on 1 November. Individual trajectories are projected on the section and shown only if less than 5 km away from it.

several PV bands present in the cloud head that are located on the outer side of the most intense positive-negative PV dipole, associated with the bent-back front and of which the SJ negative-PV filament is part. This multipole PV pattern is consistent with the results in Chagnon et al. (2013), in which a tripole PV pattern was found to be associated with a thermally direct frontal circulation (in their case associated with a single cold front), with enhanced PV at the front and reduced PV on the outer sides of the warm-sector condensational heating and cold-sector evaporative cooling. This PV pattern is also consistent with the results from the idealised simulations in Volonté et al. (2020), also showing strong PV bands located in correspondence with frontal circulations in the cloud head. The results from this section therefore suggests the presence of multiple thermally direct circulations in the cloud head, particularly intense at the bent-back front, driven by moist processes and by the associated heating and cooling.

Overall, the results presented up to this point give extensive evidence that the selected trajectories are associated with the evolution of a SJ displaying the development and subsequent release of SI (indicated by negative PV), as the SJ ascends into the cloud head and then descends and accelerates out of its tip, respectively. At the same time, intense frontal circulations and associated moist processes drive the evolution of PV in the environment around the trajectories. This case can therefore be used to investigate the diabatic processes at play during the decrease in PV along the ascending SJ, ultimately tackling the question of identifying the processes acting to make the SJ symmetrically unstable and thus enabling further acceleration during its descent.

# 3.3 Identification of diabatic processes along SJ trajectories

Figure 6 contains a box-and-whisker plot displaying the change in PV and the accumulation of the different PV tendencies in the time characterised by SJ ascent, t=-6h to t=-4h (09UTC to 11UTC).


Figure 6. Box-and-whisker plot of the total change in PV ( $\Delta$ PV) and the accumulation of individual PV tendencies ( $APV_i$ , see Table 1) along SJ trajectories from t=-6h to t=-4h (09UTC to 11UTC). Each box covers the interquartile range of the trajectory distribution, while whiskers extend to the  $5^{th}$  and  $95^{th}$  percentiles. Orange lines and green triangles indicate median and mean values, respectively.

The change in PV is shown by the leftmost box and whiskers ( $\Delta PV$ ), illustrating the generalised decrease in PV during SJ ascent, with both mean and median value close to -1.25 PVU in 2 hours. Only a small part of this PV decrease is captured by the tendencies, with mean and median of the sum all of PV tendencies ( $\sum APV_{all}$  in Figure 6) not even reaching -0.25 PVU and the extension of the whiskers pointing at the large variability across different trajectories. This large discrepancy between the actual change in PV and that captured by the accumulated PV tendencies can be ascribed to mainly two factors that must be acknowledged as limitations of this study.

- When processes are fast-changing and non-linear, as is the case with SJ dynamics, instantaneous hourly output (used in this study, meaning that the ascent is described by only three instantaneous output values for each tendency) can differ greatly from timestep-by-timestep data.
- In a small-scale, fully three-dimensional environment with steep gradients, as is the case on the cold side of the bent-back warm front, small inaccuracies in trajectory locations (favoured by too low output frequency and coarse spatial resolution) can result in large errors.







This second point explains why the agreement between PV modification and the accumulated PV tendencies becomes even worse during the final SJ descent (not shown), when small-scale processes like turbulence become more important as SI is released.

Despite these considerable limitations, it is still possible to use the individual PV tendencies to identify and describe the diabatic processes at play in the decrease of PV to negative values, i.e., the onset of SI, along the SJ. This is because, while there is large variability and little net accumulation from momentum tendencies and fast physics processes (convection and turbulence) and while radiative processes do not seem to be playing any relevant role, there is a non-negligible contribution from cloud processes. This is indicated by the sum of all the tendency accumulations associated with cloud processes ( $APV_{cloud}$ ), which is negative for most (at least three quarters) of the trajectories and with a median decrease close to -0.25 PVU and a mean decrease around -0.4 PVU.

Figure 6 contains also the boxes and whiskers associated with the individual cloud processes included in  $APV_{cloud}$ , displayed on the right side of the figure. There are four processes showing non-negligible contributions: condensation of water vapour  $(APV_{cond})$ , evaporation of cloud water  $(APV_{evc})$ , melting of ice and snow  $(APV_{melt})$  and sublimation of snow  $(APV_{subs})$ .  $APV_{cond}$  is characterised by a large variability, with most trajectories showing a PV decrease, close to -0.25 PVU for both mean and median. The cloud evaporation and melting tendency accumulations also show considerable variability, but with mean and median values near zero. The snow sublimation PV tendency accumulation is small but much better constrained, with more than 95% of trajectories experiencing a negative change in PV, despite the mean and median mean decrease only amounting to around 0.15 PVU.

Figure 7 displays the evolution over time of PV tendency accumulations traced along the SJ trajectories. We can thus use it to illustrate the evolution of the four cloud processes just mentioned throughout the evolution of the SJ, including during its ascent from t=-6h to t=-4h, to which the boxes and whiskers in Figure 6 refer.

Figure 7a shows the accumulation over time of the sum of all PV tendencies. As pointed out when describing Figure 6, only a minor part of the PV decrease during SJ ascent, exceeding 1 PVU from t=-6h to t=-4h (see also Figure 2c), is captured by the tendencies displayed in this panel. Despite the very small PV decrease during SJ ascent shown by mean and median values, the 25<sup>th</sup> percentile of tendency accumulation sees a decrease of around 1.5 PVU between t=-7h and t=-4h. This indicates that a considerable minority of trajectories does indeed experience a large PV decrease, consistent with the actual PV modification. The large differences within the trajectory set serve as reminder of the small-scale, fully three-dimensional nature of the environment near the bent-back front, where the SJ travels during its ascent and before descending off the cloud head. Figure 7b shows the PV tendency accumulation for the sum of all cloud processes. Mean and interquartile range timeseries are characterised by a general decrease of around 0.5 PVU (less for the median timeseries) between t=-5h and t=-4h, i.e., during the second part of the descent. This confirms that a substantial part of the PV decrease captured by the tendencies is caused by cloud processes. The large variability displayed from t=-7h confirms that the ascent takes place in a particularly sensitive region of the cyclone.

Looking now at the four individual cloud processes identified in Figure 6 as having the largest tendency accumulations during SJ ascent, Figures 7c and 8a show, respectively, the accumulation of PV and  $\theta$  tendencies due to condensation of water



**Figure 7.** As Figure 2 but for the accumulated PV tendencies due to (a) all diabatic processes listed in Table 1, (b) all microphysics (cloud) processes, (c) condensation of water vapour, (d) evaporation of cloud water, (e) melting of ice and snow, (f) sublimation of snow.

vapour. These timeseries show that condensation only starts having a noticeable effect when the SJ trajectories are about to start their ascent and have moved close to the bent-back warm front (Figure 4). It is at this point that most of the condensation occurs on the trajectories, as confirmed by the heating, on average exceeding 4 K, experienced during the first part of the ascent. At the same time, PV tendencies display large variability, suggesting that the trajectories are now in a highly complex environment, and the mean and median timeseries show an increase-then-decrease pattern. This decrease occurs in the second part of the ascent and is consistently shown across percentiles, indicating that is experienced by most of the trajectories.

Figures 7d and 8b show the accumulation of tendencies due to evaporation of cloud water. In this case, gradual cooling and general PV decrease are present at early stages, when the trajectories are in the boundary layer and likely below the evaporative cooling core. Afterwards, the only additional cooling, up to 1.5 K, occurs during the first part of the ascent. This cooling is not associated to any noticeable net changes in the PV median and mean timeseries, although a substantial portion of the trajectories experience changes of at least 0.5 PVU (either increasing or decreasing), again pointing at a situation where small spatial differences can lead to opposite changes in PV.



**Figure 8.** As Figure 2 but for the accumulated  $\theta$  tendencies due to (a) condensation of water vapour, (b) evaporation of cloud water, (c) melting of ice and snow, (d) sublimation of snow.

Figures 7e and 8c contain the accumulation of tendencies due to melting of ice and snow. At t=-10h they show a PV decrease that is experienced by a substantial minority of trajectories, down to near -1.5 PVU at the 5<sup>th</sup> percentile. This decrease is associated with generally small cooling for around half of the trajectories. Additional limited cooling is experience by most trajectories during their ascent, this time associated with both positive and negative changes in PV and with the median timeseries reaching the mean at around -0.5 PVU. This difference in PV modification associated with cooling due to melting is consistent with the trajectories moving from a situation of widespread cooling under the cloud head (with its maximum above the them) to a frontal environment characterised by tighter gradients.

Figures 7f and 8d show the accumulation of tendencies due to sublimation of snow. This process only takes place along the trajectories during the second part of the ascent and while it is associated with limited changes in PV, it also has very small across-trajectories variability compared to other three cloud processes considered. More than 95% of the trajectories experience a small (less than 0.5 PVU at the most) PV decrease between t=-5h and t=-4h, associated with a 1-2 K cooling. It is worth noting that both melting and sublimation of snow occur just before the end of the ascent. The cooling associated with these processes, in addition to being responsible of part of the PV decrease leading to the onset of SI, also contributes to the reduction of buoyancy leading to the start of SJ descent.

In general, it is also important to remember that fairly straightforward heating/cooling timeseries can be associated with far more complex PV modifications. This is a direct consequence of the definition of PV containing a scalar product between absolute vorticity and the gradient of  $\theta$  (Section 1). Changes in PV are more directly associated with spatial heating gradients




rather than local temperature changes along the trajectories and this is particularly evident during SJ ascent, as trajectories travel in an environment characterised by a steep front and fully three-dimensional sharp gradients in temperature and winds. In this environment, characterised by tight frontal circulations, the evolution of vorticity is also conducive to complex, multi-scale and fast-varying patterns of PV modification. Relative vorticity varies fast in the region, with its vertical component ( $\zeta_z$ ) becoming negative along the trajectories during SJ ascent and decreasing down to the point that at t=-4h also the vertical component of absolute vorticity ( $\xi_z = \zeta_z + f$ , where f indicates planetary vorticity) is negative for most trajectories (see Figure A1b, considering that at the latitudes at which this evolution is taking place f is around f in 1.1 × f 1.2 This is also consistent with the conceptual model of SJ evolution presented in Volonté et al. (2018) (see their Figs.9 and 16 and further details in Section 1) highlighting the role of tight frontal circulations in the generation of negative absolute vorticity and PV.

Therefore, to fully understand the role in the PV decrease along SJ trajectories of the different moist processes illustrated up to this point, we need to use a non-local and three-dimensional perspective, looking at the horizontal maps, vertical cross-sections and time profiles shown in Section 3.4

## 420 3.4 Identification of diabatic processes in the environment around SJ trajectories

Figure 9 contains maps of instantaneous PV tendencies for the four moist processes contributing to PV modification during SJ ascent. These maps are generated at 11 UTC (t=-4h), the end of SJ ascent, showing the tendencies at at the level nearest to the mean pressure of the trajectories (775 hPa).

Figure 9a shows the instantaneous PV tendency due to condensation of water vapour,  $PVR_{cond}$ . Large values of  $PVR_{cond}$ , both positive and negative are present throughout the whole southward extension of the warm front, mainly on its warm side, and at the bent-back warm front. Moderate positive values also cover part of the cloud head, on the cold side of the bent-back warm front. Some gaps exist between the large values at the front and the SJ trajectories lie in one of these gaps, characterised by near-zero  $PVR_{cond}$  but immediately next to an area of large negative values. These features are consistent with the results in Attinger et al. (2019, 2021), with the caveat that their composite of more than 100 rapidly developing cyclones is much less noisy than our single-case analysis of Storm Ciarán. They see condensation as a key factor behind the positive PV anomaly at the warm front, with positive values of  $PVR_{cond}$  throughout its length. Where the front bends back,  $PVR_{cond}$  shows a dipole, with negative values inside. As in our study, their results highlight the primary role of  $PVR_{cond}$  in regions characterised by ascent and indicate that the patterns of  $PVR_{cond}$  become less straightforward as frontal environments increase their curvature and heating gradients are no longer vertically oriented, such as in bent-back fronts. The PV tendency due to the evaporation of cloud water,  $PVR_{evc}$ , is displayed in Figure 9b. Its pattern is also characterised by a large number of local maxima and minima, mainly near the warm front and along its bent-back part, but also scattered around the cold sector and in the frontal-fracture region. SJ trajectories lie again in an area of negligible tendency, this time near local maxima.

Figures 9c and d contain, respectively, the instantaneous PV tendencies due to melting of snow and ice  $(PVR_{melt})$  and sublimation of snow  $(PVR_{subs})$ . Values in both panels are substantially better constrained spatially than for  $PVR_{cond}$  and  $PVR_{evc}$ , with non-negligible tendencies exclusively visible at the warm and bent-back warm fronts.  $PVR_{melt}$  is characterised by a clear along-front dipole band on the eastern part of the warm front, with positive values on the cold side, and a small

Figure 9. Maps of instantaneous PV tendencies due to (a) condensation of water vapour, (b) evaporation of cloud water, (c) melting of ice and snow, (d) sublimation of snow, indicated by the colour shading,  $\theta_e$  (magenta contours) and RH with respect to water (grey shading, > 80%) at 775 hPa and mean sea level pressure (dashed black contours) at 11UTC on 01 November. Green dots show the location of trajectories. The pressure level chosen is the closest to the mean pressure of the trajectories.

elongated positive patch covering part of the bent-back warm front. This patch is separated from the warm-front dipole by a clear gap and the SJ trajectories are located just outside it, in an area of near-zero  $PVR_{melt}$ . The primary role of melting of snow in generating an overall negative PV anomaly at the warm front is shown in Attinger et al. (2021), consistent with these results, with Attinger et al. (2019) highlighting that at the bent-back end of the front melting can also have a positive effect on

450

PV. The dipole visible here along the warm front suggests that the height of the melting core decreases going outward, thus crossing the pressure level the map refers to.

Focusing on sublimation of snow, the largest values of  $PVR_{subs}$  (Figure 9d) only reach around  $\pm 0.5$  PVU h<sup>-1</sup>, while for all the other tendencies shown in Figure 9 largest values exceed  $\pm 2.5$  PVU h<sup>-1</sup>, where the colour bar stops. These small  $PVR_{subs}$  values are located in a handful of small-scale dipole bands oriented along the warm and bent-back warm front, again with positive values on the cold side. This highlights the localised nature of PV-changing snowfall and is consistent with Attinger et al. (2019) mentioning the role of sublimation only at the bent-back end of the warm front. Despite the reduced magnitude and spatial extent of  $PVR_{subs}$  values, in this case SJ trajectories are located in area with clearly negative tendencies, while for all other tendencies shown in this figure, the trajectories were seen in an area of negligible change.

In summary, large values of  $PVR_{cond}$ ,  $PVR_{evc}$  and  $PVR_{melt}$  (with large variability and noise for the first two) are present near or along the warm and bent-back warm fronts. Values of  $PVR_{subs}$  are smaller and more localised, oriented in along-front dipoles (positive values outside) as for  $PVR_{melt}$ . However, at this time, SJ trajectories are located in an area of non-zero (negative) tendencies only for  $PVR_{subs}$ . The maps shown in Figure 9 also highlight the complexity of the bent-back front area, hinting at its three-dimensional nature and therefore stressing the importance of analysing its vertical structure. We do this by looking at the vertical sections in Figures 10-12. These sections are generated along the along-front and across-front transects shown in Figure 4e and in the maps of Figure 9, used also for the sections in Figure 5. Like all those figures, they refer to 11UTC, a time in which SJ trajectories are ascending and PV along them is decreasing fast.

The vertical cross-sections in Figure 10 show the instantaneous PV and  $\theta$  tendencies due to the condensation of water vapour ( $PVR_{cond}$ ,  $Q_{cond}$ ). The presence of large values along the front, both positive and negative, highlighted by the map in Figure 9a, is confirmed by the sections. The presence of two heating cores at the front, one at 850hPa and one centred at around 600hPa, leads to multiple positive and negative local extremes in  $PVR_{cond}$  alternating in an almost-vertical direction (Figures 10b and d). The presence of multiple additional frontal bands in the cloud head (see also Figure 5b) is associated with a heating structure with gradients in both vertical and across-front directions and thus a complex  $PVR_{cond}$  pattern.

The heating cores found in the across-front section are also visible in the along-front one (Figure 10a), in the low levels and near the cloud top, and in some places coinciding with regions particularly close to saturation. The low level heating core is associated as expected with a  $PVR_{cond}$  dipole that has negative values above the heating maximum and positive below it (Figure 10c). However, the  $PVR_{cond}$  dipole associated with the intense mid-tropospheric  $Q_{cond}$  maximum is not oriented in a negative-above-positive direction. This initially puzzling signal becomes understandable when the three-dimensional structure of condensational heating is taken into consideration.

The region of mid-tropospheric heating is tilted with height, as shown in the across-front section (Figure 10d). The area with large  $PVR_{cond}$  values along the bent-back front, shown in Figure 9a, is also tightly curved. There is thus a mix of multiple heating cores located next to each other in a fully three-dimensional pattern. In addition to this, at this time the vertical component of absolute vorticity is negative along the SJ trajectories, as explained in Section 3.3, and in the immediate surroundings, further complicating the relationship between heating and PV modification. This complex  $PVR_{cond}$  pattern is therefore inherently associated with the structure of the bent-back front and it is limited to the special location of the sections,

490

Figure 10. Vertical cross-sections of instantaneous (a,b) PV tendencies and (c,d)  $\theta$  tendencies due to condensation of water vapour (shading), RH with respect to water (light grey shading when > 80% and darker grey shading when > 98%),  $\theta_e$  (magenta contours) and freezing level (dashed brown contour) along the (a,c) AB and (b,d) CD transects shown in Figure 9, referring to 11 UTC on 1 November. Individual trajectories are projected on the section and shown only if less than 5 km away from it.

in the cloud head and just next to the front, as additional sections computed further north or south display a much simpler dipole structure in  $PVR_{cond}$  (not shown).

It is important to note that, whilst being surrounded by this complex  $PVR_{cond}$  structure, with both positive and negative extremes nearby in several directions, the SJ trajectories lie in a localised region of small negative  $PVR_{cond}$ . This further highlights the importance of the exact location in which the trajectories travel in their PV evolution and further motivates the presence of large variability in  $PVR_{cond}$  across trajectories and the large discrepancy between the actual change in PV and the sum of the time-accumulated tendencies.

Vertical sections of the tendencies due to evaporation of cloud water,  $PVR_{evc}$  and  $Q_{evc}$  show similar patterns to those of  $PVR_{cond}$  and  $Q_{cond}$  and are included in the supplementary material (Figure A2). Cooling cores are located near the top of the cloud head, and in particular of areas especially close to saturation, and at low levels. Like with the condensation tendencies, their tilted structure gives rise to a complex  $PVR_{evc}$  pattern, with SJ trajectories located in a small region characterised by near-zero tendencies whilst being surrounded by local extremes.

500

505

**Figure 11.** As Figure 10 but for (a,b) PV tendencies and (c,d)  $\theta$  tendencies due to melting of ice and snow.

Figure 11 displays vertical cross-sections of the tendencies of PV and  $\theta$  due to melting of ice and snow,  $PVR_{melt}$  and  $Q_{melt}$ . The related  $PVR_{melt}$  map, shown in Figure 9c, is characterised by a clear dipole at the warm front and a positive patch at the bent-back front, with the SJ trajectories just to the outside. The sections indicate that where cooling due to melting occurs away from frontal zones and where moist isentropes (i.e.,  $\theta_e$  contours) are near-horizontal, the associated  $PVR_{melt}$  pattern is a simple positive-above-negative dipole. The height of the cooling core and thus also of the dipole is set by the freezing level. The pattern becomes more complex closer to the bent-back front, where moist isentropes tighten and become slanted or even, locally, vertical. This is the region where the trajectories travel in during their ascent, also characterised by a dip in the inward-ascending freezing level. It is thus only in the local region of the SJ trajectories that  $PVR_{melt}$  shows a fully three-dimensional pattern.

Figure 12 shows vertical cross-sections of the tendencies of PV and  $\theta$  due to the sublimation of snow,  $PVR_{subs}$  and  $Q_{subs}$ . As already illustrated by the related map (Figure 9d), non-negligible values of  $PVR_{subs}$  are present only locally, including near the bent-back front, where the SJ trajectories are.  $\theta$  tendencies sections show that substantial sublimational cooling, in places exceeding -1 K h<sup>-1</sup> occurs on the trajectories. This is consistent with the presence of large snow water content along the trajectories at this time displayed in Figure 3d and explains the dip in freezing level co-located with the trajectories. It is worth

**Figure 12.** As Figure 10 but for (a,b) PV tendencies and (c,d)  $\theta$  tendencies due to sublimation of snow.

noting that the cooling experienced by the SJ trajectories has also the effect of reducing their buoyancy and thus favouring the start of their descent over a continuation of their ascent.

The presence of tight and slanted-to-vertical moist isentropes and the curved frontal environment in this local region lead to the associated  $PVR_{subs}$  structure to organise closer to a horizontal dipole (both in the along- and across-front directions) rather than a vertical one. The trajectories lie near the core of the negative side of this dipole. While for all the other processes examined, PV modification along the trajectories was close to zero, sublimation of snow provides a clear, albeit small, contribution to PV decrease along SJ trajectories, as they ascend in this narrow zone that just to outside the bent-back front and near the freezing level.

# 515 3.5 Time evolution of diabatic processes around SJ trajectories

In Section 3.4 we use maps and vertical cross-sections to fully characterise the three-dimensional structures of the moist processes identified in Section 3.3 as playing a role in the PV decrease along ascending SJ trajectories. However, this characterisation is performed at a single time, 11 UTC (t=-4h), near the end of the SJ ascent. To analyse the evolution in time of these processes we complement this analysis by looking at time-pressure profiles, or "Lidar plots". These mean profiles are

530

constructed by averaging the troposphere above and below the locations of the trajectories at each time, hence mimicking what would be the output of an imaginary lidar or radar scanner moving with the trajectories.

The large variability across trajectories and the especially complex structure of the tendencies associated with condensation and evaporation can lead to substantial inaccuracies and possibly misleading results in time-pressure profiles. This is particularly true as here we are using the mean location of the trajectories and instantaneous hourly data. Those profiles are provided for completeness in the supplementary material (Figure A3), while in this section of the study we focus instead on the tendencies associated with melting and sublimation of snow (Figure 13). Nonetheless, the time-pressure profiles of condensation and evaporation tendencies display useful information, such as the lowering and intensification of the mid-tropospheric condensational heating core, likely related to the descent of drier and lower- $\theta_e$  air (associated with the dry intrusion), as trajectories travel towards the bent-back front and the unique path of SJ trajectories, close but not co-located with several regions of large PV modification.

Figure 13. Time-pressure profiles ("Lidar plots", see text) of PV tendencies due to (a) melting of ice and snow and (b) sublimation of snow and of  $\theta$  tendencies due to (c) melting of ice and snow and (d) sublimation of snow. The horizontal location of the profiles is equal at each time to the horizontal location of the SJ trajectories. Grey shading indicates RH with respect to water (light when > 80% and darker when > 98%), while magenta contours indicate  $\theta_e$  and the dashed brown contour shows the freezing level.





Time-pressure profiles of PV and  $\theta$  tendencies due to melting of ice and snow (Figures 13a and c) show that until the start of their ascent the SJ trajectories travel, with no appreciable PV tendency, underneath a region of cooling and the associated positive-above-negative dipole in PV tendency. This dipole is located just below the freezing level, with the core of the cooling being at the bottom of a region particularly close to saturation. As they start ascending, the trajectories cross an area with reduced melting and then find themselves above the melting core when it increases again. This, together with the transition to the more complex  $PVR_{melt}$  structure described when looking at Figure 11, results in net cooling due to melting, shown in Figure 8c, and in generally little net PV modification but with considerable variability across trajectories (Figure 7e), caused by the several small regions of positive and negative tendency encountered during the ascent.

Time-pressure profiles of PV and  $\theta$  tendencies due to sublimation of snow (Figures 13b and d) confirm that sublimational cooling appears only in the second part of the ascent when, as explained previously and shown in Figure 3d, substantial snowfall occurs near the tip of the bent-back front and SJ trajectories reach the freezing level. The trajectories travel close to the core of the cooling, with values down to near -1 K h<sup>-1</sup> at the end of the ascent. The PV decrease is limited but non-negligible, with the mean trajectory near -0.1 PVU h<sup>-1</sup> at the same time, consistent with the pattern shown in Figures 7f, 9d and 12.

## 4 Discussion and conclusions

Sting jets (SJs) are airstreams that can form in intense extratropical cyclones and lead to exceptionally strong and damaging winds as they accelerate and descend towards the surface. There is now extensive evidence from case studies and idealised model simulations that SJ descent is often characterised by the release of (conditional or even dry) symmetric instability (SI), whose presence is indicated by negative potential vorticity (PV). The onset of this instability is driven by tight frontal circulations in the cloud head, near the bent-back warm front (Volonté et al., 2018; Clark and Gray, 2018), suggesting the likely involvement of moist processes. However, the distinct roles of individual diabatic processes have not been identified yet.

The aim of this study was therefore to identify the diabatic processes driving the onset of SI along SJs and thus enhancing the descent and acceleration of these transient mesoscale wind jets found in intense Shapiro-Keyser extratropical cyclones. To do so, we performed a model simulation of Storm Ciarán, a highly diabatic windstorm that hit the UK and western Europe on 1-2 November causing multiple fatalities and severe damage, using the ECMWF's numerical weather prediction model, the IFS. We ran the IFS using near-operational settings and adding tendencies from the individual parametrised processes to the output. In this way, having identified a SJ in this simulation and having computed Lagrangian trajectories following its path, we were able to evaluate the separate effects of individual diabatic processes on the evolution of PV and thus the onset of SI along the SJ. A primary result of this analysis is the positive answer to the question on whether diabatic tendencies can be used to identify the moist processes driving the onset of SI in SJs, although with substantial caveats. We expand on this finding and provide a point-by-point list of the key results of the study in the following.

1. The first clear result of this analysis is that our near-operational IFS simulation of Storm Ciarán contains a SJ that is associated with: a) damaging winds near the surface after its descent and b) SI (indicated by negative PV) developing earlier during its evolution.






- This SJ is consistent, in terms of spatial structure and time evolution, with those displayed by the Met Office operational forecasts illustrated in Gray and Volonté (2024) and with satellite imagery there presented. This consistency provides the necessary confidence in treating the evolution of the SJ identified in our simulation as fully plausible and realistic.
- Storm Ciarán develops multiple outward-slanted PV dipoles in its cloud head, in addition to the primary dipole associated with the bent-back warm front. The SJ becomes part of a negative-PV filament in this narrow frontal zone as it ascends while flowing towards the tip of the cloud head. The evolution of PV along the SJ, including the formation of the filament described, is consistent with what shown in previous case studies (e.g., Storm Eunice, Volonté et al. (2023b)) and idealised simulations (Volonté et al., 2020), providing further evidence that the formation of intense SJs is associated with the onset of SI (or at least one of its moist counterparts, see Section 1 and Clark and Gray (2018) for more details).
- 2. Diabatic tendencies can identify and characterise the moist processes that play a role in PV decrease and consequent onset of SI along a SJs, but caution needs to be exercised when using them, fully acknowledging their limitations. In this study, the sum of all the time-accumulated tendencies captures only a minority of the PV decrease experienced by the SJ during its ascent. This large discrepancy can be ascribed to two main factors, highlighting special characteristics and the complexity of the environment in which SJs evolve.
  - When processes are fast-changing and non-linear, instantaneous values available at frequencies that would be considered high for most synoptic-scale and cyclone-feature analyses, such as the hourly output used in this study, can still differ greatly from averages computed with data available at every model timestep.
  - In a small-scale and fully three-dimensional environment characterised by steep gradients, such as the narrow region just on the cold side of the bent-back front where the SJ ascends, small spatial inaccuracies in the calculation of trajectories can result in large errors on the values of the variables traced along them. These errors can again be caused by data frequency being not high enough and spatial resolution being not fine enough.
  - 3. Four moist processes play a non-negligible role in the PV decrease experienced by the SJ as it ascends: condensation of water vapour, evaporation of cloud water, melting of ice and snow and sublimation of snow. The first three show large variability across trajectories and, particularly for the condensation, positive and negative extremes in PV modification in the environment surrounding the trajectories. The small PV decrease caused by the sublimation of snow is instead consistent across trajectories.
    - These results, despite the limitations of a single case study, are consistent with recent literature (Attinger et al., 2019, 2021) pointing at the primary role of condensation in generating positive PV anomalies along the warm front and on the increasing importance of melting and sublimation at the bent-back end of the warm front, where the SJ ascends.





- The complex and fully three-dimensional patterns of PV tendencies caused by condensation, evaporation and melting are inherently linked with the dynamics of the bent-back front, including its outward slanted and tightly curved nature. The SJ travels in a special region in which net PV modifications from these processes, and condensation in particular, are on average limited despite the presence of large extremes nearby.
- Changes to PV caused by sublimation of snow in the environment around the SJ are mostly limited to the local region near the bent-back warm front and the tip of the cloud head, associated with substantial snowfall. The consequent small PV decrease along the ascending SJ is consistent across trajectories.
- 4. The change in PV caused by these moist processes is a consequence of the heating and cooling patterns associated with them. A direct effect of the cooling caused by melting and, particularly, sublimation of snow is the reduction in buoyancy of the SJ, which favours the start of its descent, rather than the continuation of its ascent.

In summary, in this study we use a near-operational model simulation of Storm Ciarán to illustrate the moist processes contributing to the onset of SI along a SJ and thus driving its subsequent intensification and generation of damaging winds.

We show that, despite substantial limitations in their accuracy, diabatic tendencies can be used for this purpose. Figure 14 provides a schematic representation of a key result of this analysis, i.e., the complex interplay of condensation, evaporation, melting and sublimation, all acting around the SJ as it ascends in a curved and narrow region in the cloud head, just outside the bent-back front. The fully three-dimensional structure of frontal circulations leads to complex PV tendency patterns and, excluding sublimation, the large variability across SJ trajectories. A small but clear signal of PV decrease on the SJ is ascribed to sublimation, with the associated cooling also favouring the end of SJ ascent.

This study leads naturally to further work on the role of diabatic processes, including but not limited to moist processes, in the evolution of SJs. In terms of analysis setup, the assessment of the accuracy improvements associated with having relevant variables provided or calculated at all timesteps, e.g. by using online trajectories as in Oertel et al. (2023), has the highest priority. Apart from addressing the accuracy issue, key open questions mainly concern the generalisation of the results from this study. In particular, this analysis could be extended to include:

- Events where the SJ does not show a clear ascent before starting its descent and acceleration into the frontal-fracture region, such as the SJ identified in Storm Tini (Volonté et al., 2018). In this study, the presence of an already-extended negative-PV filament near the bent-back front, of which the SJ becomes part as it completes its ascent, suggests that this behaviour could be identified even in Storm Ciarán by broadening our trajectory set. SJs that do not show an ascent from low levels before their final descent could likely spend more time in regions where the moist processes associated with bent-back front circulation cause substantial PV modification.
- Cyclones covering a range of (bent-back) frontal strengths, freezing level heights and rates of snowfall and rainfall.
   Doing so would allow us to explore the dependence of the PV modification caused by the individual moist processes examined here to different environmental conditions.

**Figure 14.** Schematic representation of the path of SJ trajectories during their ascent in the cloud head and subsequent descent (green dashed lines, with pressure indication and projection on the ground shown by green dotted lines). Green dots indicate the location of SJ trajectories near the end of their ascent. s and n coordinates are oriented along and across the direction of travel of the trajectories at the end of their ascent, with z indicating the vertical direction. Warm and bent-back warm fronts are indicated at the surface and up towards the upper-troposphere. The extent of the cloud head is indicated, together with the type of precipitation taking place, by cloud icons. Red and blue arrows indicate heating and cooling, respectively, on each side of the front. Condensation of water vapour (COND), evaporation of cloud water (EVC), melting of snow and ice (MELT) and sublimation of snow (SUBS) are indicated (in red or blue depending on whether they are associated to heating or cooling) near the locations where they are taking place.

Additional focus on the role of momentum tendencies, subject to substantial improvement in small-scale tendency accuracy. This analysis would not only cover the period of SJ ascent, but also its subsequent descent, in which moist processes play a secondary role compared to momentum-based diabatic processes.

A final implication, particularly relevant to the current warming of the North Atlantic Ocean (Kuhlbrodt et al., 2024) and thus of the environment in which these storms develop, concerns the freezing level. In this study we show the clear role of melting and particularly sublimation of snow over buoyancy and instability of the SJ. These processes are inherently tied to the height of the freezing level, which could thus exert some control on the height at which the ascent of the SJ ends and its

descent starts. This hypothesis is worth being investigated, particularly as if it is verified, then it could be also inferred that warmer storms are likely to contain SJs with more extended descent and thus, possibly, larger acceleration and higher impact.


# Appendix A: Additional figures

Figure A1. As Figure 2 but for (a) air temperature and (b) relative vorticity.

Figure A2. As Figure 10 but for (a,b)  $\theta$  tendencies and (c,d) PV tendencies due to evaporation of cloud water.

**Figure A3.** As Figure 13 but for PV tendencies due to (a) condensation of water vapour and (b) evaporation of cloud water and  $\theta$  tendencies due to (c) condensation of water vapour and (d) evaporation of cloud water.

Author contributions. AV suggested the original idea to HJ. AV and HJ conceptualised the study and acquired funding for it. MHFL performed the model simulation, including diagnostics developed by HJ and implemented by RF. AV led the analysis of the data, to which HJ and MHFL contributed substantially, later joined by RBK. AV led the paper writing, with particular sections written by HJ and MHFL. All authors provided regular feedback throughout the writing process.

*Competing interests.* AV is a member of the editorial board of Weather and Climate Dynamics. The authors declare that they have no other competing interests.

Acknowledgements. The authors would like to thank Roman Attinger and Elisa Spreitzer for their extensive discussions with AV and for the preliminary work that led to the idea of this study. The authors would also like to thank members of the Mesoscale Meteorology group at University of Reading and of the Atmospheric Dynamics group at ETH Zurich for their interest and valuable feedback on this study. This work is part of the ECMWF special project "Diabatic processes and their impact on extratropical dynamics and the hydrological cycle". We acknowledge MeteoSwiss and ECMWF for access to the ECMWF computing facilities. The work performed by AV was funded by the 2023 NCAS Visiting Scientist scheme and by the 2024 pump-priming fund of the Department of Meteorology at University of Reading. The work by MHFL was supported by an ETH Zurich Research Grant (ETH-06 21-1)

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
