# Peer review of "Identifying the diabatic processes driving the evolution of a sting jet: the case of Storm Ciarán"

_EGUsphere, 2025_

## Author Comment (AC1)

Dear Peter,

Many thanks for your message. We agree that the two reviews are comprehensive and high-quality, and we would like to thank the reviewers for their effort and for the time they spent on our manuscript.

We also agree that the key questions of:

a) multiple starting times,

b) higher resolution,

c) higher output frequency

deserve to be discussed explicitly before agreeing on the preparation of a revised manuscript. We address each question separately in the following.

**a)** Starting from the suggestion of increasing the number of start times for the Lagrangian trajectories, we agree with Reviewer 1 that this activity could be extremely interesting. We were already thinking about performing this analysis when preparing our manuscript, but in the end we decided to leave it for the future and focus on the submission. We are therefore more than willing to explore it now and here we explain how we would approach it.

The submitted manuscript includes trajectories computed from grid points with wind speed exceeding a certain threshold and retained if showing negative PV at a selected time earlier in their evolution. This approach can only be followed if SJ activity leads to the generation of a clear strong wind region, something that might not occur at earlier or later times due to the developing SJ not being able to descend and accelerate fully into the frontal-fracture. The alternative approach, that we are intending to follow, is to select trajectory start points from the narrow regions of negative PV present on the cold side of the warm and bent-back warm fronts.

For example, Figure 5e in the manuscript (also included here as Figure 1) displays the main region of negative PV. This is the region that contains the selected SJ trajectories that have just ended their ascent and are about to start descending and accelerating out of the cloud head, eventually generating the strongest 850hPa winds throughout the whole lifecycle of the storm. In addition to that region, a droplet-shaped PV< 0 area can be seen at the very tip of the cloud head and another one is instead located near the eastern part of the warm front. Given the cyclonic flow, we can treat these two additional regions as being, respectively, ahead and behind the main one.

Our potential plan would be to start backward and forward trajectories from these three regions and:

- illustrate their paths,
- assess their link to the generation of strong low-level winds,
- briefly identify the diabatic processes responsible for their formation (i.e., processes causing PV to decrease below zero).

As is also suggested by Reviewer 1, we would argue that the primary focus should still be with the PV<0 trajectories that are associated with the generation of the strongest winds, i.e., those analysed in the original submission. However, we agree that the analysis of additional starting points could have the potential to clarify some of the results obtained and increase confidence in the conclusions made. We would keep the focus of this analysis restricted to the bullet points listed above, as we believe that a more detailed investigation would result in an excessively long manuscript and would be better placed in a separate study.

[Figure]

**Figure 1** (Figure 5e in the original submission). PV (shading) and RH with respect to water (grey shading, > 80%) at the pressure levels indicated and mean sea level pressure (dashed black contours) at the times indicated. Green dots show the location of trajectories at the related times. The pressure level chosen is the closest to the mean pressure of the trajectories.

**b)** We focus now on whether running a model simulation with finer spatial resolution could help alleviating the issues with the PV budget and on why we decided to use near-operational model settings.

We indicated in the manuscript that "spatial resolution being not fine enough" could be one of the causes of the large errors in the PV tendencies, due to their role in allowing "small inaccuracies in trajectory locations" that would then lead to large errors in "small-scale, fully three-dimensional environment with steep gradients, as is the case on the cold side of the bent-back warm front". We apologise for not mentioning in that discussion that a move to finer resolution could actually worsen the errors as it would generate more noise, small-scale structures and even sharper gradients, something that Reviewer 2 also points out. In addition to this, model settings such as the convective parametrisation scheme, developed and tested in operational settings, would not necessarily be suitable for higher resolution.

Expanding on the point above, we would like to stress that the benefit of having operational setting is that we are using the model in the way it is designed, so that the likelihood of the simulation being consistent with the actual evolution of Storm Ciarán is maximised. The consistency of our IFS simulation with the operational Met Office forecasts, analysed in Gray and Volonté (2024) and found to be in agreement with observations in terms of timing and strength of the wind gusts associated with SJ descent, is a positive results for two main reasons:

- it reassures us that SJ dynamics observed in Storm Ciarán is well captured by our simulation and that the processes analysed are plausible,
- it shows that the configuration of the IFS that was operational in 2023 is capable of properly representing SJ activity in an impactful windstorm.

For the reasons just stated, we would not be inclined to re-run the IFS simulation at a finer spatial resolution even if we had the chance to do so, as the necessary amount of time and resources needed would not be justified given that associated issues could outweigh any potential gains.

Gray, S. L. and Volonté, A.: Extreme low-level wind jets in Storm Ciarán, Weather, https://doi.org/10.1002/wea.7620, 2024.

**c)** We conclude this response by addressing the comments on considering a higher time frequency for the output data of the model simulation. On this point we fully agree with both reviewers. As stated in the paper, we acknowledge that higher temporal resolution would likely benefit the study by reducing spatial inaccuracies in trajectory calculations and allowing to gain a better picture of the rapidly changing processes occurring along and around the trajectories. As Reviewer 1 notes, the lead author had

already shown the benefits on increasing time resolution when analysing SJ windstorms. Unfortunately, increasing time resolution was not possible while preparing the original manuscript and is still not possible now. This is due to the nature of the funding for this work and the technical settings available to us. Main issues revolve around the increase in the volume of the data that would be produced and the time and effort that would be required to make the new settings work, both in terms of generating sub-hourly IFS output data and of computing trajectories with Lagranto in sub-hourly mode from this particular output.

We would be happy to review the way we communicate the value of our study in its current form and to make clear what its limitations are. In particular, we would try to stress as clearly as possible the value of our methodology. It is true that the large errors in the PV budget prevent us from using the accumulation of PV tendencies along trajectories independently to identify and characterise the diabatic processes at play. However, the combined use of those timeseries with maps and vertical sections allows us to extract valuable indications on what the roles of those processes are. For example, we can see the difference between the effects of snow sublimation, limited in time and space, and those of the latent heat of condensation associated with frontal ascent, widespread but wildly varying and fully three-dimensional. We tried to explain and summarise these roles in the final cartoon schematic, but we are more than happy to review it, along with the text describing it. In general, we would like to highlight how this study can be used as proof of concept for future work that could verify and generalise its findings, particularly if taking advantage of higher time frequency.

A simple experiment that we could already perform is decreasing the speed threshold for the initial selection of SJ trajectories (e.g., from 52m/s to 50m/s), to see if widening the trajectory set to include air just outside the maximum wind core can partly mitigate the errors in the accumulated PV tendencies.

In conclusion, we hope to have provided appropriate motivation to justify how we intend to approach the key issues in the revision of our manuscript and we are happy to discuss them further. At the same time, if allowed to do so, we would like to ask for additional time to prepare our full response document and manuscript revision, mainly because of our involvement in the NAWDIC field campaign currently taking place.

With many thanks again for all the time spent reading and evaluating our work.

Best wishes

Ambrogio Volonté on behalf of all authors.